# Argipressin for prevention of blood loss during liver resection: a study protocol for a randomised, placebo-controlled, double-blinded trial (ARG-01)

Ellinor Wisén ®,[1,2] Andreas Kvarnström,[1,2] Lena Sand-Bown,[2] Magnus Rizell,[3,4] Aldina Pivodic,[5,6] Sven-Erik Ricksten,[1] Kristina Svennerholm[1,2]

For numbered affiliations see end of article.

**Correspondence to**
Dr Ellinor Wisén;
ellinor.wisen@vgregion.se

## ABSTRACT

**Introduction** Liver resection carries a high risk for extensive bleeding and need for blood transfusions, which is associated with significant negative impact on outcome. In malignant disease, the most common indication for surgery, it also includes increased risk for recurrence of cancer. Argipressin decreases liver and portal blood flow and may have the potential to reduce bleeding during liver surgery, although this has not been explored.

**Method and analysis** ARG-01 is a prospective, randomised, placebo-controlled, double-blinded study on 248 patients undergoing liver resection at Sahlgrenska University Hospital, Sweden. Patients will be randomised to one of two parallel groups, infusion of argipressin or normal saline administered peroperatively. The primary endpoint is peroperative blood loss. Secondary outcomes include need for blood transfusion, perioperative variables, length of hospital stay, the inflammatory response, organ damage markers and complications at 30 days.

**Ethics and dissemination** The study is enrolling patients since March 2022. The trial is approved by the Swedish Ethical Review Authority (Dnr 2021-03557) and the Swedish Medical Product Agency (Dnr 5.1-2021-90115). Results will be announced at scientific meetings and in international peer-reviewed journals.

**Trial registration number** ClinicalTrials.gov, NCT05293041 and EudraCT, 2021-001806-32

### STRENGTHS AND LIMITATIONS OF THIS STUDY

⇒ The study is a randomised, placebo-controlled, double-blinded trial investigating the effects of intraoperative treatment with argipressin in patients undergoing liver resection.
⇒ Broad eligibility criteria increase the generalisability of results.
⇒ Sample size is calculated from retrospective data from our study centre, and feasibility of the study protocol has been evaluated in a published pilot study.
⇒ The single centre study design may limit generalisability but decreases the influence of potential confounding factors related to clinical management.
⇒ The study participants are followed until 30 days after surgery and therefore, long-term outcome measures such as cancer recurrence and long-term mortality are not evaluated.

## INTRODUCTION

Liver resection is a central part of curative treatment in primary and metastatic liver malignancies. However, the procedure carries a high risk of extensive bleeding and need for blood transfusions. Standard anaesthetic management to prevent bleeding aims to maintain a low intraoperative central venous pressure (CVP), while surgical techniques aim to minimise blood loss by intermittent mechanical clamping of the portal vein and hepatic artery (Pringle manoeuvre, PM).[1 2] Perioperative outcome has improved due to improved surgical techniques and anaesthesiological methods, but the intraoperative risk of bleeding and need for blood transfusion during liver resection remains high. Studies indicate that 24–64% of patients undergoing liver resection require blood transfusion.[3–5] Blood loss and need for blood transfusions are independent prognostic markers for postoperative complications, tumour recurrence as well as mortality and reduced bleeding is associated with improved outcome and survival.[6 7] Therefore, it is crucial to find new interventions to decrease peroperative blood loss.

Vasopressin is an endogenous hormone with several effects including vasoconstriction. We and others have previously demonstrated that exogenous administration of vasopressin (also denoted as argipressin or arginin-vasopressin) has the capacity to decrease portal and hepatic blood flow in patients undergoing liver surgery.[8–10]



Furthermore, treatment with vasopressin led to reduced bleeding and improved outcome in an experimental pig model of blunt liver trauma.[11 12] A recent randomised controlled trial presented evidence indicating that argipressin decreased transfusion requirements in bleeding trauma patients, without evidence of side effects.[13] These findings indicate that argipressin may be a promising candidate to reduce blood loss during liver resection, although this has not been investigated.

Systemic inflammation caused by the surgical procedure is associated with an increased risk of tumour recurrence and negative outcome.[14] Argipressin has anti-inflammatory properties, and we hypothesise that argipressin reduces systemic inflammation caused by hepatic surgery.[15 16]

During major surgery most patients need an infusion of a pressor agent with vasoconstrictive properties, usually norepinephrine. Vasoconstrictive drugs, including argipressin and norepinephrine, confer a risk of reduced perfusion of internal organs. There is a knowledge gap concerning the use of argipressin during liver resection in conjunction with a low CVP strategy. Thus, the third part of this study will evaluate whether argipressin in this setting will affect levels of organ damage biomarkers compared with placebo.

We hypothesise that treatment with argipressin reduces bleeding and transfusion requirements during liver resection. The hypotheses will be tested in a randomised, placebo-controlled, double-blinded trial, defined ARG-01.

## OBJECTIVES

The primary objective is to test whether treatment with argipressin reduces blood loss during liver resection. Secondary objectives include assessment of argipressin's effects on the need for blood transfusions, as well as on the inflammatory response. Exploration of potential effects on kidney, cardiac and intestinal tissue biomarkers will be performed. The duration of the surgery and the resection phase as well as use of PM, in addition to radicality of resection will be determined. Peroperative haemodynamics, use of vasopressor and achievement of low CVP goals will be evaluated. Finally, total length of stay in hospital and postoperative complications at 30 days are noted.

## METHODS AND ANALYSIS
### Study design

ARG-01 is a single-centre, prospective, randomised, placebo-controlled, double-blinded, 30-day follow-up trial that will include 248 patients undergoing liver resection at Sahlgrenska University Hospital, in Gothenburg, Sweden. Inclusion was started in March 2022 and the last participant is estimated to be recruited at the end of 2024. The study protocol and its feasibility are based on two previous physiological studies and a prospective pilot trial performed by our research group at Sahlgrenska

University Hospital.[9 10 17] In a subgroup of 88 patients, the effects of argipressin on the inflammatory response will be evaluated.

The study is performed in adherence with the Standard Protocol Items: Recommendations for Interventional Trials (SPIRIT) guidelines. The complete study protocol and the SPIRIT checklist are attached as supplements (online supplemental files 1 and 2, respectively). The study design is illustrated in figure 1.

### Screening and recruitment

Patients are designated to open or laparoscopic surgery at the surgeon's discretion. All patients planned for liver resection are noted on a screening list. Written information about the study is sent to the patients before the preoperative anaesthesia consultation, where screening for eligibility will take place. All patients fulfilling inclusion criteria and without exclusion criteria will be invited to participate. Following oral and written consent, patients are assigned a subject number in the electronic Case Report Form (eCRF) Research Electronic Data Capture (REDCap), which will be used for data collection throughout the study.

### Inclusion criteria
► Patients scheduled for liver resection (open or laparoscopic).
► Age ≥18 years.
► American Society of Anaesthesiologists class I–III.
► Signed informed consent

### Exclusion criteria
► Participant does not understand the given information, and/or cannot give written informed consent.
► Simultaneous operation of tumour with other localisation, or surgery for superficial single hepatic tumour less than 2 cm, expected to be of short duration and with minimal blood loss.
► Terminal kidney failure (preoperative estimated glomerular filtration rate <15 mL/min).
► Pregnancy or lactation.
► Known allergy to Empressin.
► Patient included in other interventional study, interacting with the endpoints in the present study, or previous randomisation in this study.
► Hyponatraemia (S-Na <130 mmol/L).
► Patient considered ineligible for other surgical or medical reason.
► Present infection.

Patients with systemic inflammatory disease, inflammatory bowel disease or preoperative corticosteroid treatment will not be eligible for subgroup analysis.

### Stratification, randomisation and blinding

Stratification according to open or laparoscopic surgery and planned size of resection ('small' vs 'large') is performed in REDCap. A 'small' resection is defined as maximum four wedge resections or maximum two segmental resections. All resections more extensive than

**ARG-01 Flow Diagram**

**Figure 1** ARG-01 flow chart. ITT, intention-to-treat.

'small', including right or left lobe resections, are defined as 'large'. Laparoscopic surgery is exclusively performed on 'small' resections.

Before randomisation, eligibility for subgroup participation is considered, depending on logistical factors (immediate sample handling) and exclusion criteria for subgroup analysis. In total 88 patients will be included in the subgroup analysis. Equal numbers of patients from treatment and placebo group will be recruited, and an equal distribution between open and laparoscopic surgery will be ensured according to stratification at randomisation.

On the day of surgery patients will be randomised to one of two parallel groups, treatment (argipressin) or placebo (normal saline), at a ratio of 1:1 using the eCRF computer-based randomisation tool. The randomisation is performed by an unblinded, assisting nurse who is not involved in patient care or in study-specific activities, and with REDCap access limited to the randomisation tool.

The nurse prepares the blinded study drug according to randomisation. The study drug syringe is provided to

the anaesthetic nurse in charge of the patient, hence both patient and all healthcare providers involved in patient care are blinded to its content. Argipressin and normal saline are both clear and colourless fluids. REDCap generates a randomisation number which is linked to a sealed envelope with information about the allocation. The envelope is used only in case unblinding is crucial for patient safety, that is, if an unexpected serious event jeopardising the safety of the subject is occurring. Allocation will not be revealed to the research group until the database is locked.

### Study drug
Patients are randomised to receive a continuous infusion of argipressin (Empressin) 0.8 U/mL, or placebo (normal saline, NaCl 9 mg/mL), at a constant rate of 0.056 mL/kg/hour. The infusion is started after induction of anaesthesia and placement of a central venous catheter (CVC) and continued until the end of surgery. The dose of argipressin equals 0.045 U/kg/hour corresponding to 3.6 U/h for a patient of 80 kg body weight.

The dose of argipressin in the present study is based on the dose of vasopressin used in two large randomised controlled trials, VANISH and VANCS, where the effect of norepinephrine and vasopressin were evaluated in patients with septic shock and in vasoplegic shock after cardiac surgery, respectively.[18][19] This dose is approximately 50% higher than the one used in a randomised controlled trial on trauma patients with haemorrhagic shock, where vasopressin decreased the requirement for blood transfusion, without evidence of side effects.[13] Furthermore, the vasopressin infusion in the mentioned study was given for 48 hours which is a significantly longer duration than in our study protocol. The dose regime in the present study protocol is related to body weight, in contrast to studies by others, where a constant infusion of argipressin (U/h) is often used.[13][18][19]

### Interventions

All study interventions are described in detail in table 1 (online supplemental file 3).

Intraoperative blood loss will be recorded at the end of surgery. The volume of blood transfusion during surgery and at postoperative day (POD) 1, and POD 2 and 5, respectively, will be recorded. Baseline blood samples are obtained before induction of anaesthesia, and subsequent samples will be collected until discharge from hospital, but not later than POD 2 for laparoscopic surgery and POD 5 for open surgery. Analysis of plasma levels of white blood cell count (WBC), platelets, haemoglobin, creatinine, C-reactive protein (CRP), albumin, interleukin (IL)-1$\beta$, IL-6, IL-8, IL-10, monocyte chemoattractant protein (MCP)-1, stromal cell derived factor (SDF)-1$\alpha$, intracellular adhesion molecule (ICAM), complement C3a, C5b-9, high sensitive-Troponine-I (hs-TnI), intestinal fatty acid binding protein (I-FABP), urine-creatinine and urine (TIMP-2) × (IGFBP-7) (Nephrocheck®) will be performed.

Duration of surgery, resection phase and total time with applied PM are noted. Achievement of low CVP-goal and the cumulative dose of norephinephrine are registered at the end of surgery, as well as urine output and the use of diuretics at POD 1. Haemodynamic measurements are recorded intraoperatively. Follow-up data at 30 days will be obtained from the SweLiv registry. Background data including Clinical Frailty Scale are obtained at the preoperative consultation and from medical charts.

Anaesthesia, perioperative monitoring and treatment and postoperative care will be performed according to the clinical routine for liver resection at Sahlgrenska University Hospital. An epidural catheter is placed and activated during surgery for open surgery patients. An arterial line and CVC are placed in all patients. Cardiac output monitoring (PICCO®) is used at the discretion of the anaesthesiologist. To achieve a low CVP (defined as CVP <5 mm Hg or a reduction of baseline CVP with 33%) a restrictive fluid regime, infusion of a crystalloid at a rate of 1.5 mL/kg/hour throughout the peroperative phase, is used. Diuretics, low positive end-expiratory pressure and/

or nitroglycerin is used if necessary to achieve low CVP. Norepinephrine or other inotropes are used as needed to maintain adequate haemodynamic goals.

Surgical aspects of liver resection, including use of PM, is managed by the surgeon according to clinical routine. The use of Surgicel® or other haemostatic agents such as tranexamic acid are documented. Conversion of laparoscopic surgery to open resection is noted in the eCRF. If the surgery is aborted, due to disseminated malignancy or other reasons, the patient is withdrawn from the study and no more protocol-specific interventions are performed.

## OUTCOMES
### Primary outcome

The primary outcome is the volume of blood loss at the end of surgery, measured according to a standardised protocol. Fluid in suction canisters is measured and the quantities of irrigation fluid and ascites are subtracted. Gauze pieces and sponges are wrung before being counted and the fluid is then suctioned into the canisters.

### Secondary outcomes

Secondary outcomes include the following:
- The proportion of patients receiving blood transfusions and the volume of blood transfusions during surgery and hospitalisation (but not longer than POD 2 for laparoscopic patients and POD 5 for open surgery patients).
- Inflammatory response: change in levels of WBC, CRP, platelets and albumin, IL-1$\beta$, IL-6, IL-8, IL-10, MCP-1, SDF-1$\alpha$, ICAM, C3a, C5b-9 from baseline, to end of surgery and POD 1 and 2.
- Duration of PM (total minutes), resection phase and surgery and use of tranexamic acid.
- Achievement of CVP goal, total dose of norepinephrine used during surgery, total urine output and total dose of furosemide at POD 1.
- Postoperative complications documented at POD 30. Radicality of resection.
- Length of stay in hospital.
- Change of levels of plasma creatinine from baseline to POD 1, 2 and 5. Urine creatinine and urine (TIMP-2) × (IGFBP-7) (Nephrocheck) from baseline to 3 hours after end of surgery.
- Change of plasma hs-TnI levels from baseline to end of surgery and POD 1.
- Change of plasma lactate and I-FABP levels from baseline to end of surgery, at 3 hours after surgery and POD 1.

## DATA MANAGEMENT

Data from medical charts, anaesthesia and transfusions records (including assessment of blood loss), laboratory results and data from the SweLiv registry is recorded for each patient in REDCap. Source data is handled according to Good Clinical Practice (GCP) and the General Data Protection Regulation (GDPR).

## MONITORING

The study monitoring is performed by Gothia Forum, Gothenburg, Sweden, to ensure that data is complete and correctly registered, consistent with the source data, in addition to reviewing compliance of the study protocol, laws and regulations. Blinded and unblinded monitors are assigned to the project. Monitoring is performed according to the study monitoring plan. Informed consent and primary outcome are monitored for all included patients, as well as study drug compliance. The complete source data monitoring rate is 35%, which may be adjusted after future monitoring reports.

This low-risk phase IV study does not have a Data Safety Monitoring Board. The study group closely monitors all patients and assesses all potential adverse events (AE) and serious AE (SAE).

## ADVERSE EVENTS AND SAFETY

AE and SAE, defined as an undesired medical event occurring during the study period (until POD 2 for laparoscopic surgery and POD 5 for open surgery), are noted and assessed for causality. Definition of AE and SAE criteria are described in detail in the study protocol (online supplemental file 1). The complication rate after liver resection varies between 30–60% and most SAE:s are expected to be related to surgery and not to the study drug.[20] All AE data will be coded and listed individually and reported in future publications. In case a suspected unexpected serious adverse reaction occurs, it will be reported by the sponsor to the European Database Eudra-Vigilance and to the Ethics Committee. The study drug can be discontinued at any time during surgery, in case of a suspected adverse reaction, and unblinding may be performed as described above.

## Sample size

Calculation of sample size was performed based on data from open and laparoscopic liver resections performed at Sahlgrenska University Hospital during the period January 2019 to October 2020. Log-transformation of the blood volume variable normal distribution was applied. The sample size calculation was performed using a two-sided t-test on the log-transformed variable that will correspond to the selected method for the primary analysis, that is, generalised linear model with lognormal distribution. To be able to find at least 35% reduction in the mean blood loss in the treatment arm compared with the placebo arm (962 mL vs 625 mL, respectively), that is, a difference of at least 0.43 in the log-scale assuming SD of 1.17, with a level of significance of 5%, and a power of 80%, 118 patients are needed to be included in each arm. Assuming a dropout rate of 5%, 248 patients in total will need to be included, 124 patients in each arm. A limited number of patients will be discovered fulfilling the exclusion criteria after randomisation and after start of surgery. Since the investigational product is given from the start of surgery these patients will be part of the safety population and their outcome regarding survival will be collected at the end of the study (30 days). In these cases, surgery will be immediately stopped, and they will be excluded from the study and the modified intention-to-treat (mITT) population, that is, all efficacy analyses. To account for these dropouts 248 mITT evaluable patients will be included, but not more than 272 randomised patients in total.

For the subgroup analysis 44 patients in the argipressin and placebo arm, respectively, will be included (total n=88). The number of patients selected for extended inflammatory testing is based on a pilot study performed by Wisén *et al* ('Myocardial, renal and intestinal injury in liver resection surgery-A prospective observational pilot study'; data regarding inflammatory response was generated, but not published), performed at Sahlgrenska University Hospital.[17]

## STATISTICAL ANALYSIS PLAN

The statistical analysis plan (SAP), including detailed information of all planned analyses, will be finalised, and signed off before the database lock and unblinding (online supplemental file 4). Any changes made in the SAP after the database lock will be noted as post-hoc changes and new analyses as post-hoc analyses.

### Study populations

#### Modified intention-to-treat population

The mITT population will include all patients that were randomised into the study fulfilling all inclusion criteria and none of the exclusion criteria. The mITT analyses will be performed on an as-randomised basis.

#### Per protocol population

The per-protocol (PP) population will include all patients that were randomised into the study and that do not have any major protocol violations in the study affecting the outcome. The final PP population will be defined in a blinded manner, after the clean file of the study database and before the database lock and unblinding of the study treatment. PP analyses will be performed on an as-treated basis. Major protocol violations will be collected during the study and may include following but is not limited to:

► Excessive bleeding not related to the resection phase, such as injury of vena cava.

#### Safety population

The safety population will include all patients that were randomised into the study and that have undergone study intervention with either argipressin or placebo. Safety analyses will be performed on an as-treated basis.

### General methodology

All analyses will be performed using SAS software V.9.4 or later (SAS Institute, Cary, North Carolina, USA).

Descriptively, continuous variables will be described by mean, SD, median, IQR and range as appropriate, and categorical variables by number and percentage.

For test between two groups Fisher's exact test will be used for dichotomous variables, Mantel-Haenszel $\chi^2$ trend test for ordered categorical variables, $\chi^2$ test for non-ordered categorical variables and two-sample t-test or Mann-Whitney U test for continuous variables as appropriate based on their distribution.

## Primary endpoint

The primary endpoint, blood loss at the end of surgery, will be evaluated on the mITT population comparing all eligible patients treated with argipressin versus placebo, on 0.05 significance level. The primary analysis will be performed using a generalised linear model with lognormal distribution, adjusted for randomisation strata of laparoscopic, open small and open large surgery. Relative risk (RR) with 95% CI and p value will be described. Diagnostic plots will be reviewed to assure that the model assumptions are fulfilled. Graphically, the primary endpoint will be described using boxplots. Analysis performed on the PP population will serve as robustness analysis.

## Secondary endpoints

The main secondary analysis will be performed on the mITT population and robustness analysis on the PP population. Continuous secondary endpoints will be studied using general linear model adjusted for laparoscopic, open small and open large surgeries, if the model assumptions are shown to be fulfilled. Generalised linear model with lognormal or Poisson distribution will be used, similarly adjusted, if the model assumptions are shown to be satisfied for these distributions instead. Otherwise, nonparametric unadjusted Mann-Whitney U test will be used. For dichotomous secondary endpoints, such as need for transfusion, binary logistic regression will be applied, adjusted for the randomisation strata as above.

Time-to-event data, such as length of stay in hospital, will be graphically described using Kaplan-Meier technique and analysed using Cox regression model. Assumption of proportional hazard will be checked by studying interactions with time in study.

Difference in least square means, RR, OR and HR with 95% CI between the treatment groups and associated p values will be the main measures presented.

Changes in inflammatory laboratory data and in organ damage markers before and after surgery at different time points will be analysed using mixed models for repeated measures using normal or other distribution as appropriate for the studied data. Covariance matrix will be selected as the one with the lowest Akaike's information criterion, among unstructured, compound symmetry, autoregressive and Toeplitz.

Tests for IL and cytokine levels (IL-1b, IL-6, IL-8, IL-10, MCP-1, SDF-1a, ICAM, C3a, C5b-9) will only be analysed in a subgroup of 44 patients in each parallel group (total n=88). The results will be presented as explorative data.

The secondary endpoints that will be included in the confirmatory fixed sequential testing per the order below

following the confirmation of the primary analysis, with 0.05 significance level, will be:
► The volume of blood transfusions during surgery and hospitalisation.
► The proportion of patients receiving blood transfusion.

All other secondary endpoints will be evaluated in exploratory manner, and hence not adjusted for multiplicity. All exploratory tests will be cautiously considered, taking into account the magnitude and clinical relevance of the results.

No interim analysis is planned. Data will be analysed by the statistician before being presented to the research group.

## Subgroup analyses

The effect of treatment on the primary endpoint, blood loss and on the need for transfusion will be studied for following subgroups:
► Surgical technique (open vs laparoscopic liver resection).
► Extent of surgery (small vs large resection).
► Extent of surgery within open surgery (small vs large resection).
► Indication for surgery (divided into one to two largest categories vs others).
► Hepatic cirrhosis (yes vs no).
► Age ($\leq$ median vs > median).

For a limited number of patients surgical technique might change during the surgical procedure from laparoscopic to open. In the subgroup analysis the actual surgical technique will be used. The final list of subgroups will be specified in the SAP. The analyses will include investigation of the interaction between the intervention and the considered variable using the same regression methods as specified above for blood loss and need for transfusion. The results will be graphically described in forest plots.

## PATIENT AND PUBLIC INVOLVEMENT STATEMENT

Patients have not been involved in designing this study. Although, if this study demonstrates that argipressin has the capacity to decrease blood loss and/or decrease requirements for transfusion compared with placebo, the study may lead to improved care and outcome for patients undergoing surgical treatment due to malignant tumours or metastases within the liver. Argipressin is an approved and well-known drug used in clinical practice within the field of anaesthesia and intensive care. If argipressin is beneficial for the patient, treatment can be readily used in clinical practice in Sweden as well as globally, in a large cohort of patients within the near future. However, a future, larger multicentre trial would further strengthen any results from this study.

## ETHICS AND DISSEMINATION

The study, including patient information and informed consent form, have been approved by the Swedish Ethical

Review Authority (Dnr 2021-03557) and the Swedish Medical Product Agency (Dnr 5.1-2021-90115). Written informed consent will be obtained from all participants before any study specific procedures are performed.

The study is conducted in compliance with the International Council for Harmonisation of Technical Requirements for Pharmaceuticals for Human Use, GCP, applicable national regulatory requirements (Swedish Medical Products Agency regulations 2011:19) and in accordance with the ethical principles in the Declaration of Helsinki (2013).

We aim to use the Swedish National Database (SND, https://snd.gu.se/en) for meta data sharing, and to make pseudonymised data available on reasonable request. The study results will be shared at local and international scientific meetings and published in international peer-reviewed scientific journal(s) regardless of the results.

**Author affiliations**
[1]Department of Anaesthesiology and Intensive Care, Institute of Clinical Sciences, Sahlgrenska Academy, University of Gothenburg, Gothenburg, Västra Götaland, Sweden
[2]Department of Anaesthesia and Intensive Care, Sahlgrenska University Hospital, Gothenburg, Sweden
[3]Transplantation Center, Sahlgrenska University Hospital, Gothenburg, Sweden
[4]Deparment of Surgery, University of Gothenburg Institute of Clinical Sciences, Goteborg, Västra Götaland, Sweden
[5]Department of Clinical Neuroscience, Institute of Neuroscience and Physiology, Sahlgrenska Academy, University of Gothenburg, Gothenburg, Sweden
[6]APNC Sweden, Gothenburg, Sweden

**Acknowledgements** The authors would like to acknowledge Ingrid Eiving and Marita Ahlqvist for their expert help with blood sample collection and data management support.

**Contributors** KS is the principal investigator for the trial, and EW principal co-investigator. KS and EW are responsible for study design, preparation and drafting of study protocol, implementation of study procedures, involved in creation of the statistical analysis plan (SAP) and wrote the manuscript. S-ER, AK, MR and LS-B are involved in study design, study protocol and reviewed and edited the manuscript. AP drafted and prepared the SAP, sample size calculation and reviewed and edited the manuscript. All authors revised and approved the final version of the manuscript.

**Funding** This work was supported by the Department of Anaesthesia and Intensive Care, Sahlgrenska University Hospital, and the following grants: Gothenburg University, and the Sahlgrenska University Hospital (ALF-LUA), Gothenburg Medical Society (GLS-974156, GLS-961705), Sahlgrenska University Funds (SU-961353), Anna-Lisa Bror Björnsson Foundation, Tore Nilson's Foundation, The Foundation for Transplantation and Cancer Research.

**Competing interests** None declared.

**Patient and public involvement** Patients and/or the public were not involved in the design, or conduct, or reporting, or dissemination plans of this research.

**Patient consent for publication** Not applicable.

**Provenance and peer review** Not commissioned; externally peer reviewed.

**ORCID iD**
Ellinor Wisén http://orcid.org/0000-0003-3198-7162

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
