## [Reviewer comments · BMJ Open]

ARTICLE DETAILS

TITLE (PROVISIONAL)	Argipressin for prevention of blood loss during liver resection; a study protocol for a randomised placebo-controlled, double-blinded trial (ARG-01).
AUTHORS	Wisén, Ellinor; Kvarnström, Andreas; Sand-Brown, Lena; Rizell, Magnus; Pivodic, Aldina; Ricksten, Sven-Erik; Svennerholm, Kristina

VERSION 1 – REVIEW

REVIEWER	Martel, G Ottawa Hospital Research Institute, Surgery
REVIEW RETURNED	04-May-2023

GENERAL COMMENTS	Thank you for asking me to review this manuscript by Wisen and colleagues. The authors present a protocol for a single-center 242-patient RCT of argipressin vs placebo in liver resection. The primary outcome is blood loss. The trial has been open to accrual since March 2022. The protocol is very well written and complete. The investigators have included a trial statistician and the statistical plan is sound. I have very few criticisms. I have included minor comments and questions to improve clarity for the reader: -The authors should clarify the difference between argipressin and vasopressin for the reader. They should further justify their choice of argipressin, rather than vasopressin, given that the latter was used in some of the trials they have used to inform their study design and drug dosages.-It would be valuable for the authors to further define what is meant by other interventional studies with endpoints interacting with the current trial (exclusion criteria).-Please define what co-interventions to minimize blood loss will be allowable in this trial and how these will be accounted for in the analysis (e.g. cell saver, TXA, Pringle, etc.).-How do the authors propose to visually assess blood loss from sponges? These should be weighed. I would also recommend using a blood loss estimation formula on day 2 after surgery to use as a sensitivity analysis, as this is likely to be more reproducibly accurate than visual estimation (e.g. Flordal or Ross equation).-The choice of the primary endpoint is the only major criticism I have. The study intervention may very well be effective, but blood loss is such a poor outcome that the trial may end up null or worse difficult to interpret.-Please justify or clarify the limitation of blood transfusion assessment to 2 days vs 5 days for lap and open cases. Surely, if someone has a complicated lap operation and needs a transfusion on day 3, this should be accounted for in the transfusion outcome. Placing different caps for lap and open is concerning. Moreover,
---

	whatever cap is used should reflect the probability that postop transfusions are related to intraoperative blood loss. I would thus argue that transfusions should be monitored to at least one week or ideally 30 days. -I would recommend describing the primary efficacy analysis as a modified ITT given the post-randomization exclusion of open-close patients. -I do not agree with the contention that major bleeding outside of the liver resection phase should be considered a protocol deviation. This is a rare but occasionally expected consequence of liver surgery, and perhaps more importantly, the investigational product is given for the whole surgery (not just the liver transection phase). -The strata included in the analysis are lap, open small and open large. I am assuming that this implies there is no lap large group, but this should be stated explicitly.
--	---

REVIEWER	Karanicolas, Paul Sunnybrook Health Sciences Centre, Surgery
REVIEW RETURNED	12-May-2023

GENERAL COMMENTS	This protocol describes a randomized, placebo-controlled, blinded trial comparing argipressin to placebo in patients undergoing liver resection. Primary outcome is perioperative blood loss with secondary outcomes including blood transfusion and perioperative outcomes. Sample size is 248 patients at a single hospital. The trial began enrolling patients in March 2022. The trial is described by the authors as pragmatic and large. The rationale for this trial is strong and overall the design is sound. I do have several specific points that could be clarified in the manuscript:  1. I would not describe the study as large, in fact it is likely under-powered for many of the key secondary outcomes. It is also limited by the fact that it is single-centre, so most of the anaesthetic and surgical practices are likely standardized, and not representative of the range of practices. It is difficult for a single-centre trial to truly be pragmatic. 2. Participants will be followed until 30 days after surgery. There is strong evidence that many post-operative adverse events occur between days 30-90 after surgery. Why not follow patients at least until post-operative day 90? 3. Although the trial is described as pragmatic, there are several elements that are much more explanatory. For example, the outcome of estimated blood loss is not a patient –important outcome. Nor is the inclusion of several mechanistic key secondary outcomes. The authors might want to complete the PRECIS tool (https://www.precis-2.org/) to assess the extent of pragmatism on all domains and ensure that the design elements meet their intentions. 4. The definition of “small” and “large” liver resections seem quite arbitrary and open to interpretation. The authors should justify this or consider using a validated measure that is associated with the outcomes of the trial – for example the three point transfusion score (Br J Surg. 2017 Mar;104(4):434-442. doi: 10.1002/bjs.10416.) 5. Do all patients undergoing liver resection at this site routinely have central venous catheters placed? Many centers have moved away from this practice, which may limit the generalizability of the findings from this trial.
---

	6. Are other co-interventions allowed – for example, can patients receive tranexamic acid? What about cell-salvage – is this used at all at this center? 7. Estimated blood loss is notoriously difficult to measure and very subjective. Thus it is less associated with adverse events compared to blood transfusion. There should be strong justification for selection of this as the primary outcome rather than blood transfusion or another patient-important outcome. Further, the method for estimating blood loss should be explicitly stated and monitored – for example, will gauze/sponges be weighed? Currently the protocol describes “visual assessment”; what does this mean? 8. Why are blood transfusions only measured to POD 2 in laparoscopic patients and POD 5 in open surgery patients? Similarly why are adverse events only measured during this timeframe? Why does it differ between laparoscopic and open surgery? 9. What is the justification for the reduction of 35% in mean blood loss used in the sample size calculation? 10. The manuscript would be strengthened by a description of the knowledge translation strategy.
--	--

VERSION 1 – AUTHOR RESPONSE

Reviewer: 1

Dr. G Martel, Ottawa Hospital Research Institute

Comments to the Author:

Thank you for asking me to review this manuscript by Wisen and colleagues. The authors present a protocol for a single-center 242-patient RCT of argipressin vs placebo in liver resection. The primary outcome is blood loss. The trial has been open to accrual since March 2022. The protocol is very well written and complete. The investigators have included a trial statistician and the statistical plan is sound. I have very few criticisms. I have included minor comments and questions to improve clarity for the reader:

-The authors should clarify the difference between argipressin and vasopressin for the reader. They should further justify their choice of argipressin, rather than vasopressin, given that the latter was used in some of the trials they have used to inform their study design and drug dosages.

Answer: The endogenous substance vasopressin (also called argipressin or arginin-vasopressin) is available as a synthetic peptide. The active substance of the drug is argipressin or vasopressin used synonymously. Argipressin (Empressin®) is the only substance currently available in Sweden, and is the drug used in clinical practice at our center. This has been clarified in the manuscript (page 5).

-It would be valuable for the authors to further define what is meant by other interventional studies with endpoints interacting with the current trial (exclusion criteria).

Answer: Inclusion in any concomitant ongoing trials where the study protocol includes pharmaceutical or other interventions aiming to impact the perioperative process, could potentially influence both primary and secondary outcomes of our study. The study is performed at a teaching hospital, with several active research groups. For example, patients included in a pharmacokinetic study of a traceamount of chemotherapy administered perioperatively were excluded from inclusion (although interaction with study endpoints seem unlikely).

-Please define what co-interventions to minimize blood loss will be allowable in this trial and how these will be accounted for in the analysis (e.g. cell saver, TXA, Pringle, etc.).

Answer: As stated in the protocol, all surgical and anesthesiologic methods considered beneficial by the treating physician are allowed. This is clarified in the manuscript (page 9 and 10). The use of Pringle maneuver is allowed, and surgeons are encouraged to evaluate the need of its use carefully. Total minutes of use of Pringles maneuver will be documented in the eCRF and reported. The use of TXA has now been added to the protocol and eCRF and will be analyzed. Cell saver is not common practice during cancer surgery at our center and is therefore not used in this trial. For oozing surface after resection, human fibrinogen / thrombin bandage may be used at the end of resection.

-How do the authors propose to visually assess blood loss from sponges? These should be weighed. I would also recommend using a blood loss estimation formula on day 2 after surgery to use as a sensitivity analysis, as this is likely to be more reproducibly accurate than visual estimation (e.g. Flordal or Ross equation).

Answer: We agree that blood loss may be a complex variable to quantify. In the present study the measurement of blood loss is performed using a standardized procedure by our research nurses, assessing suction canisters with marked volumes (ml), subtracting irrigation fluids and ascites. The clinical routine at our operation theater is that all sponges are wet before use, and then wrung before counting, with the excess fluid suctioned up into the canisters. Based on unpublished data from a pilot study performed at our center, where all sponges were weighed, we concluded that the weight of the sponge or gauze before and after use did not differ substantially. If any sponge is not wrung before counting, a pre-specified volume is added to the estimated blood loss. Hence, the term "visual assessment" is not fully accurate. We appreciate the feedback and have clarified this in the manuscript (page 9). The assessment is performed by the same blinded study nurses in both randomised groups, and the randomised, blinded design of the study will ensure equal measurement approach in both groups.

The suggestion to use a formula for calculation of blood loss was discussed in the research team when designing the trial. As stated in the review article by Tran et al 2020, there is no gold standard for blood loss estimation (1). According to Tran, visual techniques tend to underestimate blood loss, while formulas tend to render higher values. This difference was not statistically significant in pooled analysis. Formulas often assume that the hematocrit of the shed blood is constant, while in clinical practice blood loss is often replaced by crystalloids or colloids, hence diluting the blood, which may contribute to this discrepancy (2).

We struggle to find a formula applicable in our setting, where a substantial proportion of study participants receive blood products during surgery, which is not accounted for in the Ross formula. The Flordal formula accounts for the transfusion of packed red blood cell units, but not for the fact that not all losses are replaced by red blood cells alone after the "allowed" or calculated blood loss is reached. In our practice, further transfusions with both crystalloids and colloids other than red cells such as plasma or albumin may be used. At day 2 several factors such as fluid status, further transfusions and oral intake may make the formulas unreliable.

-The choice of the primary endpoint is the only major criticism I have. The study intervention may very well be effective, but blood loss is such a poor outcome that the trial may end up null or worse difficult to interpret.

Answer: We thank the reviewers for highlighting this subject, which has been discussed at length among the authors. By today, when more than half of the participants have been enrolled, it is not possible to change the primary endpoint.

This study was designed to test the effect of argipressin on blood loss. Blood loss reflects the immediate perioperative effects of argipressin in contrast to transfusion of blood products, that may be influenced by several other factors such as preoperative anemia, especially after a few days. If a variable like blood transfusion would have been chosen as primary outcome, the possible effect of argipressin wouldn't be detected in patients with small amount of blood loss (not in need of transfusions). However, we acknowledge that transfusion is also of great importance for the individual patient, and need of blood transfusion and transfused volume are therefore reported as the main secondary outcomes.

The power calculation was based on data on estimated blood loss at our center, and the present study is designed according to this number of participants.

-Please justify or clarify the limitation of blood transfusion assessment to 2 days vs 5 days for lap and open cases. Surely, if someone has a complicated lap operation and needs a transfusion on day 3, this should be accounted for in the transfusion outcome. Placing different caps for lap and open is concerning. Moreover, whatever cap is used should reflect the probability that postop transfusions are related to intraoperative blood loss. I would thus argue that transfusions should be monitored to at least one week or ideally 30 days.

Answer: At our tertiary center, the majority of patients are discharged after 2 or 5 days respectively. Of the ones requiring further care, many are discharged to other hospitals in the region. We aim to monitor the immediate effects of the short-acting drug argipressin during surgery, and transfusions given after the expected time for recovery may depend on other factors than the primary blood loss at surgery. Transfusions will be expressed as event rate adjusted for number of follow up days (rather than frequency and number of transfusions), meaning that adjustment will be made for varying follow-up time among study patients.

-I would recommend describing the primary efficacy analysis as a modified ITT given the post-randomization exclusion of open-close patients.

Answer: We appreciate this suggestion and will refer to the primary efficacy analysis as modified ITT (mITT) in the manuscript (page 12).

-I do not agree with the contention that major bleeding outside of the liver resection phase should be considered a protocol deviation. This is a rare but occasionally expected consequence of liver surgery, and perhaps more importantly, the investigational product is given for the whole surgery (not just the liver transection phase).

Answer: Major, unexpected blood loss from for example a rift in the inferior caval vein does influence our outcome substantially, but it does not reflect the effect of our study drug. We do not consider it a standard event even though we are aware that it is a possible surgical complication. Patients subjected to this type of complication will be handled as per the Statistical Analysis Plan, i. e. included in the modified ITT and safety analyses but excluded from the per protocol analysis.

-The strata included in the analysis are lap, open small and open large. I am assuming that this implies there is no lap large group, but this should be stated explicitly.

Answer: Yes, that is correct and this is specified in the manuscript (page 7).

Reviewer: 2

Dr. Paul Karanicolas, Sunnybrook Health Sciences Centre

Comments to the Author:

This protocol describes a randomized, placebo-controlled, blinded trial comparing argipressin to placebo in patients undergoing liver resection. Primary outcome is perioperative blood loss with secondary outcomes including blood transfusion and perioperative outcomes. Sample size is 248 patients at a single hospital. The trial began enrolling patients in March 2022. The trial is described by the authors as pragmatic and large.

The rationale for this trial is strong and overall the design is sound. I do have several specific points that could be clarified in the manuscript:

1. I would not describe the study as large, in fact it is likely under-powered for many of the key secondary outcomes. It is also limited by the fact that it is single-centre, so most of the anaesthetic and surgical practices are likely standardized, and not representative of the range of practices. It is difficult for a single-centre trial to truly be pragmatic.

Answer: Thank you for this comment. We realize that we have not used the word “pragmatic” in the strict scientific sense, and this is adjusted in the manuscript. The power calculation is based on data on estimated blood loss at our center, and the study is designed according to this number of participants (248 evaluable patients). The power calculation was performed for the primary variable as per the standard procedure, and not for all secondary outcomes. Most of the secondary outcomes will be reported descriptively, except for the volume of blood transfusions during surgery and hospitalization, and the proportion of patients receiving blood transfusion that will be included in the sequential testing procedure. When designing the study protocol, initiation of a multi-center trial was discussed. However, since this is a first clinical evaluation of argipressin’s effect during liver resection and patient enrollment is proceeding according to plan, a single-center trial was considered a reasonable first step. In this early phase, control over surgical and anesthesiologic factors makes the study results less susceptible to confounding factors.

2. Participants will be followed until 30 days after surgery. There is strong evidence that many post-operative adverse events occur between days 30-90 after surgery. Why not follow patients at least until post-operative day 90?

Answer: We aim to monitor the immediate effects of argipressin during surgery. The clinical routine in Sweden is a follow-up visit after 30 days, and the results from that visit are documented in the SweLiv registry (<https://statistik.incanet.se/SweLiv/>). Any patient suffering an adverse event during the expected postoperative course (2 or 5 days) will be followed until the event is resolved, and all adverse events will be reported.

3. Although the trial is described as pragmatic, there are several elements that are much more explanatory. For example, the outcome of estimated blood loss is not a patient –important outcome. Nor is the inclusion of several mechanistic Key secondary outcomes. The authors might want to complete the PRECIS tool (<https://www.precis-2.org/>) to assess the extent of pragmatism on all domains and ensure that the design elements meet their intentions.

Answer: The study description “pragmatic” was pointed out in previous comments as inadequate, and we appreciated this being brought to our notice. In the revised version, the wording is adjusted to better describe our study design.

4. The definition of “small” and “large” liver resections seem quite arbitrary and open to interpretation. The authors should justify this or consider using a validated measure that is associated with the

outcomes of the trial – for example the three point transfusion score (Br J Surg. 2017 Mar;104(4):434-442. doi: 10.1002/bjs.10416.)

Answer: This definition of “small” and “large” is used in other studies by members in the study group and show correlation to risk of complications, even if the “large” group includes resections that are not traditionally regarded as such (3). An important point in the present study is to assess the impact of argipressin in “small” resections compared to the larger resections, and to evaluate differences between different types of resections within and between study groups in a subgroup analysis.

5. Do all patients undergoing liver resection at this site routinely have central venous catheters placed? Many centers have moved away from this practice, which may limit the generalizability of the findings from this trial.

Answer: Yes, as described in the protocol, central venous catheters are placed in all patients undergoing liver surgery, as this is routine practice in Scandinavia.

6. Are other co-interventions allowed – for example, can patients receive tranexamic acid? What about cell-salvage – is this used at all at this center?

Answer: As stated in the protocol, all surgical and anesthesiologic methods considered beneficial by the treating physician are allowed. This is clarified in the manuscript (page 9 and 10). The use of Pringle maneuver is allowed, and surgeons are encouraged to evaluate the need of its use carefully. Total minutes of use of Pringles maneuver will be documented in the eCRF and reported. The use of TXA has now been added to the protocol and eCRF and will be analyzed. Cell saver is not common practice during cancer surgery at our center and is therefore not used in this trial. For oozing surface after resection, human fibrinogen / thrombin bandage may be used at the end of resection.

7. Estimated blood loss is notoriously difficult to measure and very subjective. Thus it is less associated with adverse events compared to blood transfusion. There should be strong justification for selection of this as the primary outcome rather than blood transfusion or another patient-important outcome. Further, the method for estimating blood loss should be explicitly stated and monitored – for example, will gauze/sponges be weighed? Currently the protocol describes “visual assessment”; what does this mean?

Answer: We thank the reviewers for highlighting this subject, which has been discussed at length among the authors. By today, when more than half of the participants have been enrolled, unfortunately, it is not possible to change the primary endpoint.

This study was designed to test the effect of argipressin on blood loss. Blood loss reflects the immediate perioperative effects of argipressin in contrast to transfusion of blood products, that may be influenced by several other factors such as preoperative anemia, especially after a few days. If a variable like blood transfusion would have been chosen as primary outcome, the possible effect of argipressin wouldn't be detected in patients with small amount of blood loss (not in need of transfusions). However, we acknowledge that transfusion is also of great importance for the individual patient, and need of blood transfusion and transfused volume are therefore reported as the main secondary outcomes.

Answer: We agree that blood loss may be a complex variable to quantify. In the present study the measurement of blood loss is performed using a standardized procedure by our research nurses, assessing suction canisters with marked volumes (ml), subtracting irrigation fluids and ascites. The clinical routine at our operation theater is that all sponges are wet before use, and then wrung before counting, with the excess fluid suctioned up into the canisters. Based on unpublished data from a pilot

study performed at our center, where all sponges were weighed, we concluded that the weight of the sponge or gauze before and after use did not differ substantially. If any sponge is not wrung before counting, a pre-specified volume is added to the estimated blood loss. Hence, the term “visual assessment” is not wholly accurate—we appreciate the feedback and have clarified this in the manuscript (page 9). The assessment is performed by the same blinded study nurses in both randomised groups, and the design of the study will ensure equal potential error in both groups.

8. Why are blood transfusions only measured to POD 2 in laparoscopic patients and POD 5 in open surgery patients? Similarly why are adverse events only measured during this timeframe? Why does it differ between laparoscopic and open surgery?

Answer: We aim to monitor the immediate effects of the short-acting drug argipressin during surgery, not to monitor overall postoperative complications rates. Blood transfusions given after the expected time for recovery may depend on other factors than the primary blood loss at surgery. Transfusions will be expressed as event rate adjusted for number of follow up days (rather than frequency and number of transfusions), meaning that the metric will be adjusted for different follow-up time among patients. At our tertiary center the majority of patients are discharged after 2 or 5 days respectively. Of the ones requiring further care, many are discharged to other hospitals in the region, where transfusion regimes may differ. Adverse events occurring after the expected time frame of recovery does not monitor the immediate effects of argipressin but rather the postoperative complication rate, which we also assess at the 30-day follow up. However, adverse events will be followed until resolution, regardless of time needed.

9. What is the justification for the reduction of 35% in mean blood loss used in the sample size calculation?

Answer: A 35% relative reduction of the mean blood loss at our center corresponds approximately to one unit of bank blood, which we considered clinically relevant.

10. The manuscript would be strengthened by a description of the knowledge translation strategy.

Answer: Argipressin is a well-established drug used in clinical practice, which means that a positive study outcome could be readily translated into change of clinical routine. A future larger, multi-center trial would further strengthen the results. These comments are added in the manuscript (page 14).

1. Tran A, Heuser J, Ramsay T, Mclsaac DI, Martel G. Techniques for blood loss estimation in major non-cardiac surgery: a systematic review and meta-analysis. *Can J Anaesth.* 2021;68(2):245-55.
2. Gross JB. Estimating allowable blood loss: corrected for dilution. *Anesthesiology.* 1983;58(3):277-80.
3. Scherman P, Syk I, Holmberg E, Naredi P, Rizell M. Risk Factors for Postoperative Complications Following Resection of Colorectal Liver Metastases and the Impact on Long-Term Survival: A Population-Based National Cohort Study. *World J Surg.* 2023.

VERSION 2 – REVIEW

REVIEWER	Martel, G Ottawa Hospital Research Institute, Surgery
REVIEW RETURNED	26-Jul-2023

GENERAL COMMENTS	Thank you for addressing my questions. The methodological choices have been adequately argued and justified.
REVIEWER	Karanicolas, Paul Sunnybrook Health Sciences Centre, Surgery
REVIEW RETURNED	28-Jul-2023
GENERAL COMMENTS	Thank you for the revisions and congratulations on conducting this clinical trial, I look forward to seeing the results.